# Adsorption Properties of Comb-Shaped Polycarboxylate Dispersant onto Different Crystal Pyraclostrobin Particle Surfaces

**DOI:** 10.3390/molecules25235637

**Published:** 2020-11-30

**Authors:** Liying Wang, Chong Gao, Jianguo Feng, Yong Xu, Danqi Li, Lixin Zhang

**Affiliations:** 1Institute of Functional Molecules, Shenyang University of Chemical Technology, Shenyang 110142, China; 15002499849@163.com (C.G.); lidanqi@yeah.net (D.L.); 2School of Horticulture and Plant Protection, Yangzhou University, Yangzhou 225009, China; jgfeng@yzu.edu.cn; 3Innovation Center of Pesticide Research, Department of Applied Chemistry, College of Science, China Agricultural University, Beijing 100193, China; cauxy@cau.edu.cn

**Keywords:** pyraclostrobin, crystal form, polycarboxylate dispersant, stability, adsorption

## Abstract

The stability of the suspension system of the two crystal forms of pyraclostrobin is evaluated using multiple light technology, and the adsorption performance of polycarboxylate dispersant on the surface of two different crystalline pyraclostrobin particles is compared in combination with XRD, FTIR, XPS, and SEM from the microscopic view. The adsorption kinetics and thermodynamics studies of 2700 on the surfaces of different crystalline forms of pyraclostrobin particles show that the adsorption process of 2700 on the surfaces of pyraclostrobin crystal forms II and IV conform to pseudo-second-order kinetic adsorption model. The *E*a values for crystal forms II and IV are 12.93 and 14.39 kJ∙mol^−1^, respectively, which indicates that both adsorption processes are physical adsorption. The adsorption models of 2700 on the surfaces of pyraclostrobin crystal forms II and IV are in accordance with Langmuir adsorption isotherms. The **∆***G*_ad_ values of crystal forms II and IV are negative and the **∆***S*_ad_ values are positive at different temperatures. Therefore, the adsorption processes are spontaneous and accompanied by entropy increase. The results of this study provide an important theoretical basis for the selection of polycarboxylate dispersants in the suspension of pyraclostrobin. This study also provides a reference for the research of polycrystalline pesticide suspension concentrate.

## 1. Introduction

Suspension concentrate, which is a development direction of modern pesticide formulations, is one of the four environmentally friendly formulations recommended by FAO. It has increasingly become a major research area in water-based formulations due to its high efficiency, environmental compatibility, safe use, and convenient application. Suspension concentrate disperses the insoluble or poorly soluble solid drugs in water into water with a certain degree of fineness to form a uniform and stable suspension system [1]. Suspension concentrate is a heterogeneous solid-liquid dispersion system between colloidal dispersion and coarse dispersion, and it is thermodynamically and kinetically unstable [2]. Problems, such as delamination, pasting, bottoming, and crystal growth, often occur in the production, storage, and transportation; these issues restrict the application and development of suspension concentrate. Research on the physical stability and stability mechanism of the suspension concentrate is the key to solving a series of problems.

In the preparation process of the suspension concentrate, the growth of crystals and the coalescence or flocculation between particles can be inhibited by adding a suitable dispersant, to improve the physical stability of the suspension concentrate [3,4]. In recent years, many researchers studied the physical stability of suspension concentrate from the microscopic level, particularly the adsorption capacity, adsorption force, and adsorption morphology of the dispersant on the surface of the original drug particles [5]. The adsorption thermodynamics and adsorption kinetics have also been used to explore the effect of the dispersant [6]. For example, Ma et al. performed infrared spectroscopy, X-ray photoelectron spectroscopy, and scanning electron microscopy to study the adsorption performance of polycarboxylate dispersant on the surface of fipronil particles, and combined them with adsorption theory to explore the stability mechanism of polycarboxylate dispersant on suspension concentrate [7]. Hao et al. used the oscillation adsorption method to determine the adsorption performance of various anionic dispersants on the surface of imidacloprid particles and examined the effects of the types and amounts of dispersants on the stability of imidacloprid suspension concentrate [8,9,10]. These studies provide theoretical basis and technical support for improving the stability of pesticide suspension concentrate.

The material crystal form is the manifestation of material existence. Different crystal forms of substances can directly affect the physical and chemical properties of substances. Different crystal forms of the same substance may have significant differences in solubility, melting point, and dissolution [11]. The emergence of pyraclostrobin, boscalid, and other crystal forms patents in recent years has gradually increased the attention to the influence of pesticide crystal forms on the stability of pesticide suspension concentrate. Pyraclostrobin is a strobilurin fungicide with pyrazole structure and is one of the most widely used fungicides in the world. It also has a wide bactericidal spectrum, high efficiency, and low toxicity [12]. Pyraclostrobin is a representative polycrystalline compound in pesticide. It has four crystal forms, and two of them exist stably (form II and IV). In the practical production of pyraclostrobin suspension concentrate, the product is often pasting, layering, and bottoming caused by the crystal form of pyraclostrobin. The different crystal forms of the pyraclostrobin can directly affect the physical stability of the pyraclostrobin suspension concentrate. Therefore, the stability of different crystal suspension systems of the same pesticide should be explored.

Polycarboxylate dispersant is an anionic dispersant that is commonly used in pesticide suspension concentrate, and it has good dispersing properties [13,14]. It uses electrostatic repulsion and steric hindrance to prevent the aggregation of ultrafine solid particles in the suspension system and improve the stability of suspension concentrate [15]. This study uses two crystal forms of pyraclostrobin and polycarboxylate dispersant as models, and the stability of the suspension system of the two crystal forms of pyraclostrobin and polycarboxylate is evaluated using multiple light technology in combination with XRD, FTIR, XPS, and SEM to compare the adsorption performance of polycarboxylate dispersant on the surface of two different crystalline pyraclostrobin particles from the microscopic view. At the same time, the stability mechanism of the two crystalline pyraclostrobin suspension systems is further discussed in consideration of adsorption thermodynamics and kinetics, and the theoretical basis for the preparation of stable suspension concentrate for polymorphic pesticides is provided.

## 2. Results and Discussion

### 2.1. Stability Analysis

The multiple light scatterometer uses near-infrared light as the light source and reflects the change in the particle size and concentration of the sample through the curve of the transmitted light and backscattered light with time. We analyzed the stability of the sample by calculating the TSI [16]. Notably, a larger TSI, means the system is more unstable.

Figure 1 shows the time-dependent spectrum of transmitted and backscattered lights of two crystalline pyraclostrobin suspension systems. The left and right sides of the scan graph represent the bottom and top of the sample cell, respectively. The change in backscattered light intensity on both sides is related to the migration of particles (precipitation, floating, or clarification). The middle of the scan represents the middle of the sample cell, and the change in backscattered light intensity reflects the change in particle size (agglomeration or flocculation).

The figure shows that the backscattered light intensity between samples A and B do not change over time, which indicates that the particle size of the solid particles in the two crystalline pyraclostrobin suspension systems do not change, and the particles do not agglomerate. Obvious transmitted light appears in sample A, which implies that the sample of pyraclostrobin crystal form II suspension system is stratified.

From the TSI value of Table 1, we obtain that the TSI (top) of pyraclostrobin crystal form II is 34.6, and the TSI (top) of pyraclostrobin crystal form IV is 19.9. Therefore, the stratification phenomenon of the former is more obvious than that of the latter. The TSI (total) of pyraclostrobin crystal form II is 8.3, and the TSI (total) of pyraclostrobin crystal form IV is 5.1. A smaller TSI, indicates that the system is more stable. Therefore, the suspension system of pyraclostrobin crystal form IV is more stable than that of pyraclostrobin crystal form II suspension system.

### 2.2. Crystal Structure Analysis

XRD is one of the main methods to study the polymorphs of drugs [17]. The diffraction patterns of polymorphs are different from each other and have characteristic peaks, which are especially suitable for crystal type discrimination. XRD analysis reveals the two pyraclostrobin crystal and their samples absorbing 2700. As shown in Figure 2, the reflection peaks in the X-ray powder diffraction graph of pyraclostrobin crystal form II are the same as those after the adsorption of 2700, both of which contain its five characteristic peaks. The reflection peaks in the X-ray powder diffraction graph of pyraclostrobin crystal form IV are the same as those after the adsorption of 2700, both of which contain all of its five characteristic peaks. Crystal forms II and IV of pyraclostrobin adsorb the dispersant 2700, and the crystalline forms do not change.

The crystal form of the drug can be converted to each other under the conditions of the external environment. The above mentioned XRD analysis results imply that the polycarboxylate dispersant 2700 has no effect on the two crystal forms of pyraclostrobin in the suspension system of the two crystal forms of pyraclostrobin.

### 2.3. Adsorption Morphology and Adsorption Type Analysis

#### 2.3.1. Particle Microscopic Surface Morphology Analysis

SEM was conducted to study the microscopic morphology of the two crystalline pyraclostrobin particles before and after the adsorption of dispersant 2700. Figure 3 shows that before adsorbing the dispersant, the surfaces of two pyraclostrobin crystal forms are smooth and the surface of crystal form IV is much smoother than that of crystal form II. After adsorbing 2700, several small particles are absorbed to the smooth surface of pyraclostrobin particles and distributed orderly. This result is due to the fact that the hydrophobic groups of 2700 coat the surface of pyraclostrobin particles and their hydrophilic groups are fully exposed. These conditions effectively prevent the agglomeration of pyraclostrobin particles and improve the stability of solid–liquid dispersion system.

#### 2.3.2. Adsorption Type of Dispersant on Particle Surface Analysis

In this experiment, the adsorption force of dispersant 2700 on the surface of different crystalline particles of pyraclostrobin was analyzed by FTIR. The results are shown in Figure 4.

As shown in Figure 4, no new absorption peak appears after pyraclostrobin crystal form II or IV adsorbs the dispersant 2700. In the FTIR spectrum, the absorption peak dose not move significantly to the low or high frequency range after adsorbing dispersant 2700. No hydrogen or chemical bond occurs between the dispersant 2700 and pyraclostrobin crystal form II or IV. 

#### 2.3.3. Adsorption Model of Dispersant on Particle Surface

The adsorption model can more intuitively describe the adsorption process of dispersant 2700 on the surface of pyraclostrobin particles (Scheme 1). The dispersant 2700 is adsorbed on the pyraclostrobin particles surface through the long chain of the hydrophobic skeleton and mainly van der Waals force. At the same time, the carboxylate anion group on the side chain is hydrophilic and can be free in the water and form a certain thickness of adsorption layer on the surface of the pyraclostrobin particles.

### 2.4. Adsorption Thermodynamics

#### 2.4.1. Adsorption Isotherms

Adsorption is a surface phenomenon, which is a process in which adsorbates gather from the solvent to the surface of the adsorbent. Adsorption equilibrium is controlled by factors such as the properties of the adsorbent and the adsorbate, the composition of the solution, and the temperature [18]. In this experiment, dispersant 2700 is adsorbate, pyraclostrobin crystal forms II and IV are adsorbents, and the solvent is water. The adsorption isotherms of dispersant 2700 on the surface of pyraclostrobin crystal form II and IV particles at different temperatures are shown in Figure 5.

Figure 5A shows that the adsorption capacity of 2700 on the surface of pyraclostrobin crystal form II increases with the increase in equilibrium concentration at the same temperature. At a low equilibrium concentration, the adsorption capacity increases rapidly, and it slows down and ultimately reaches the saturation adsorption when the equilibrium concentration increases to a certain extent. At different temperatures, the adsorption capacity of 2700 on the surface of pyraclostrobin crystal form II increases with the rise in temperature.

As shown in Figure 5B, the adsorption capacity of 2700 on the surface of pyraclostrobin crystal form IV increases with the increase in equilibrium concentration at the same temperature. When the equilibrium concentration increases to a certain extent, the increase rate slows down and the adsorption capacity ultimately reaches the saturation adsorption. At different temperatures, the adsorption capacity of 2700 on the surface of pyraclostrobin crystal form IV decreases with the rise in temperature.

#### 2.4.2. Adsorption Models

The adsorption isotherms of dispersant 2700 on the surface of pyraclostrobin crystal form II and IV particles at different temperatures are of type I (the type of adsorption isotherm referred to in this article comes from the IUPAC name). Therefore, the Langmuir and Freundlich adsorption equations are used to fit the data, respectively. The fitting results are shown in Table 2.

As shown in Table 2, the adsorption isotherms of 2700 on the surfaces of the different crystalline forms of pyraclostrobin particles are highly fitted with the Langmuir model. Specifically, the correlation coefficients (R^2^) are over 0.97 and 0.98 for crystal forms II and IV, respectively. With the increase in temperature, the adsorption capacity of 2700 on the surface of pyraclostrobin crystal form II increases from 1.613 mg·g^−1^ to 2.561 mg·g^−1^, while the adsorption capacity on the surface of pyraclostrobin crystal form IV decreases from 2.018 mg·g^−1^ to 1.559 mg·g^−1^. The difference in the saturated adsorption capacity of 2700 on the surface of pyraclostrobin crystal forms II and IV becomes more significant with the increase in temperature.

For the suspension system with 2700 as the dispersant, pyraclostrobin crystal form II requires a larger amount of dispersant to achieve adsorption equilibrium than pyraclostrobin crystal form IV with the increase in temperature.

#### 2.4.3. Adsorption Thermodynamics Analysis

Through the study of adsorption thermodynamics, we can understand the trend, degree and driving force of adsorption process, which is important to explain adsorption characteristics, laws, and adsorption mechanism.

Table 3 shows that the adsorption free energy **∆***G*_ad_ values of pyraclostrobin crystal forms II and IV at different temperatures are negative and the absolute value increases with the rise in temperature. Therefore, the adsorption processes of 2700 on the surfaces of two pyraclostrobin crystal forms are spontaneous. The adsorption enthalpy change **∆***H*_ad_ of pyraclostrobin crystal form II is positive, which means that the adsorption process is an endothermic process. The adsorption enthalpy change **∆***H*_ad_ of pyraclostrobin crystal form IV is negative, which means that the adsorption process is an exothermic process. The adsorption entropy change **∆***S*_ad_ of pyraclostrobin crystal forms II and IV at different temperatures are positive, which implies that the adsorption processes of 2700 on the surfaces of two pyraclostrobin crystal forms are processes of entropy increment. In the whole adsorption processes, the movement of 2700 on the surfaces of two pyraclostrobin crystal forms becomes increasingly chaotic. The adsorption of 2700 on two pyraclostrobin crystal forms will decrease the entropy, and the desorption of water will increase the entropy at the same time. Furthermore, the molecule volume of 2700 is much larger than that of water, and the adsorption of 2700 on the surfaces of two pyraclostrobin crystal forms is accompanied by the desorption of multiple water molecules. Therefore, the entropy increase caused by the desorption of water molecules on the surfaces of two pyraclostrobin crystal forms is much larger than the entropy decrease caused by the adsorption of 2700. This phenomenon results in processes of entropy increment.

The type of adsorption force is decided by the numerical range of the adsorption enthalpy change. The chemical bond force is greater than 60 kJ·mol^−^^1^, the hydrogen bond force is 20–40 kJ·mol^−1^, and the van der Waals force is less than 20 kJ·mol^−1^. As shown in Table 3, the absolute values of the adsorption enthalpy change of pyraclostrobin crystal forms II and IV are both lower than 20 kJ·mol^−1^, which means that the main adsorption force of 2700 on the surfaces of two pyraclostrobin crystal forms belongs to van der Waals force.

### 2.5. Adsorption Layer Thickness Analysis

The dispersant adsorbed on the surface of the solid particles can form an adsorption layer with certain thickness to maintain the stability of solid–liquid dispersion system, and determining the adsorbed layer thickness can visually reflect the steric effect. In this study, XPS was used to determine the adsorption layer thickness of dispersant 2700 on the surfaces of two pyraclostrobin crystal forms at different temperatures.

Figure 6A,B that the electron peak intensities of N and Cl are weakened, while the electron peak intensities of C and O are significantly enhanced. At the same time, a new electronic peak of Na appears after the pyraclostrobin crystal form II adsorbs dispersant 2700. This result is due to the coating effect of the dispersant, which implies that the dispersant 2700 forms a good adsorption layer on the surface of the pyraclostrobin crystal form II particles (Table 4).

As shown in Figure 6C,D, the electron peak intensities of N and Cl are weakened, while the electron peak intensities of C and O are significantly enhanced. At the same time, a new electronic peak of Na appears after the pyraclostrobin crystal form IV adsorbs dispersant 2700. This finding is due to the coating effect of the dispersant, which indicates that the dispersant 2700 forms a good adsorption layer on the surface of the pyraclostrobin crystal form IV particles (Table 5).

The dispersant 2700 does not contain Cl element and pyraclostrobin contains Cl element. Thus, Cl element is selected as the characteristic element. By measuring the attenuation degree of the Cl 2p photoelectron after passing the adsorbed layer (Figure 7), the average adsorbed layer thickness can be approximately calculated according to Equations (14) and (15).

Table 6 shows that 2700 forms an adsorption layer with certain thickness on the surface of two pyraclostrobin crystal forms. The adsorbed layer thickness of 2700 on the surface of crystal form II increases with the rise in temperature, while the adsorbed layer thickness of 2700 on the surface of crystal form IV decreases with the rise in temperature. This finding is in accordance with the results of adsorption isotherms and thermodynamics analysis.

### 2.6. Adsorption Kinetic

#### 2.6.1. Adsorption Kinetic Curves

The adsorption rate is an important indicator to evaluate the speed that dispersants reach the adsorption equilibrium on the surfaces of the pesticide particles. To investigate the effect of temperature on the adsorption rate of 2700 on the surfaces of pyraclostrobin crystal forms II and IV, the relationships between the adsorption capacity and the adsorption time of 2700 on the surfaces of two pyraclostrobin crystal forms at different temperatures were determined (Figure 8).

Figure 8 show that the adsorption capacity of 2700 on the surfaces of two pyraclostrobin crystal forms increases as adsorption time progresses at different temperatures. At the initial stage of adsorption, the adsorption capacity increases quickly. Then, the curve becomes gentle, and the adsorption capacity eventually reaches the adsorption equilibrium over time. The faster adsorption rate can be attributed to the higher concentration of 2700 in the solution than that on the surface of particles at the initial stage. According to the chemical equilibrium principle, the adsorption is carried out positively and the adsorption rate is faster. During adsorption, the concentration of 2700 on the surfaces of particles increases and the adsorption rate slows down until the adsorption equilibrium.

#### 2.6.2. Adsorption Kinetic Models

The adsorption rate constant is an important evaluation index for the adsorption process. Two kinds of adsorption kinetics models were used to study the adsorption process of 2700 on the surface of two pyraclostrobin crystal forms at different temperatures (Figure 9).

The fitting results in Table 7 show that the correlation coefficients R^2^ of the pseudo-second-order kinetic equation are all more than 0.999, which indicates that the pseudo-second-order kinetic model is most suitable to fit the adsorption process. The pseudo-second-order kinetic model contains the whole adsorption process. Thus, it can reflect the adsorption kinetics of 2700 on the surfaces of two pyraclostrobin crystal forms realistically and comprehensively.

The apparent activation energy *E*a is calculated by the rate constant *k*_2_ according to Equation (14). For the adsorption process of 2700 on the surface of pyraclostrobin crystal form II, the fitting equation is y = −1555x + 3.5691, the correlation coefficient is 0.974, and the *E*a is 12.93 kJ∙mol^−1^ calculated from the slope. For the adsorption process of 2700 on the surface of pyraclostrobin crystal form IV, the fitting equation is y = −1730.5x + 3.9641, the correlation coefficient is 0.999, and the *E*a is 14.39 kJ∙mol^−1^ calculated from the slope. The physical adsorption *E*a is usually 5–40 kJ∙mol^−1^, and the chemical adsorption *E*a is generally 40–800 kJ∙mol^−1^. Thus, the adsorption processes of 2700 on the surfaces of two pyraclostrobin crystal forms all belong to physical adsorption as supported by consisting with FTIR analysis results.

## 3. Materials and Methods

### 3.1. Materials and Reagents

Pyraclostrobin (crystal IV) with a purity of 98% was provided by Noposion Agrochemicals, Co., Ltd. (Shenzhen, China) and pyraclostrobin (crystal II) with the same purity was made by our laboratory. The molecular structure of pyraclostrobin is shown in Scheme 2A. Two pyraclostrobin crystal and their respective samples after adsorbing the dispersant were characterized by XRD (SmartLab, Rigaku, Japan).

Polycarboxylate (TERSPERSE^®^2700, abbreviated as 2700, Mw = 7808) was provided by HUNTSMAN (Woodlands, TX, USA), the structure of which is shown in Scheme 2B. Analytical grade methanol was purchased from Sinopharm Chemical Regent Co., Ltd. (Beijing, China) and the distilled water was self-made.

### 3.2. Stability Analysis

We adopted wet grinding process, weighed a certain amount of pyraclostrobin and polycarboxylate dispersant, made up to 100% with deionized water, and made all the materials wet and flowable. We used a high-shear emulsifier to cut for 1–2 min and poured the sheared sample into a sand milling cylinder. We added a certain amount of zirconia beads for 3 h grinding, and the resultant discharged to obtain the sample.

We shook the prepared pyraclostrobin suspension system (20 mL) into a glass sample bottle, scanned it every 1 h with a multiple light scatterometer (Turbiscan Tower, Formulaction, France) at 25 °C for 48 h [19,20]. The relative stability of each system can be compared by the turbiscan stability index (TSI). The calculation formula of the stability kinetic index is given as follows:(1)$$\mathrm{TSI}=\frac{\sum_{h}\left|{\mathrm{scan}}_{i}\left(h\right)-{\mathrm{scan}}_{i-1}\left(h\right)\right|}{H},$$
where i is the number of measurements, h is the scanning height of the instrument, scan is the intensity of backscattered or transmitted light, and H is the maximum height of the measurement.

### 3.3. Microscopic Analysis

Two pyraclostrobin crystals and their respective samples after adsorbing the dispersant were dried. Then, the dried samples were observed by SEM (S-4800, Hitachi, Kanto, Japan) after being sputtered with a thin layer of gold.

### 3.4. FTIR Analysis

The suspensions of pyraclostrobin crystal II and IV after being oscillated at 298.15 K were centrifuged, and the precipitation was dried at vacuum condition and then measured by FTIR (Tracer-IR 100, Shimadzu, Osaka, Japan) over potassium bromide pellets.

### 3.5. Adsorption Thermodynamic

The oscillating adsorption method was adopted to determine the adsorption isotherms of 2700 on the surfaces of two crystal of pyraclostrobin particles [21,22]. Briefly, a certain amount of pyraclostrobin was placed in a 100 mL flask. Then, a certain concentration of 2700 solution was added. The flasks were sealed and placed at 298.15, 308.15, and 318.15 K for a certain period of time under oscillation condition. When the adsorption reached equilibrium, the supernatant was filtered through a filter with a pore size of 0.22 μm. Then, the filtrate was diluted to the appropriate concentration, and its mass concentration was determined using an ultraviolet visible spectrophotometer (UV-1800, Shimadzu, Osaka, Japan). At the same time, a blank experiment was performed to eliminate the interference of pyraclostrobin in the water on UV absorption. The adsorption amount is calculated according to the following equation:(2)$$Q_{t}=\left(C_{0}-C_{t}+C_{b}\right)V/m,$$
where *Q_t_* is mass per unit mass of the drug adsorption dispersant (mg·g^−1^), *C*_0_ is the mass concentration of the dispersant solution (mg·L^−1^), *C_t_* is the mass concentration of the solution after adsorption equilibrium (mg·L^−1^), *C_b_* is the mass concentration of blank sample (mg·L^−^^1^), *V* is volume of solution (L), *m* is the mass of pyraclostrobin (g).

Different adsorption curves reflect different adsorption methods. Different adsorption models are used to fit the adsorption isotherms to obtain the relevant adsorption coefficients [23].

1.Langmuir equation

The Langmuir equation has a great general range of application [24,25]. Despite its simplifying assumptions, which may appear unrealistic in most of the applications, it is popular mainly because of its simple form and reasonably good predication characteristics. Equations (3)–(5) present a general form of the Langmuir equation.
(3)$$Q_{t}=aC_{e}Q_{e}/\left(1+aC_{e}\right),$$
(4)$$Q_{t}/Q_{e}=aC_{e}/\left(1+aC_{e}\right),$$
(5)$$C_{e}/Q_{t}=1/aQ_{e}+C_{e}/Q_{e},$$

2.Freundlich equation

This equation is a classical isotherm based on the assumption of logarithmic reduction in the heat of adsorption by increasing surface coverage [26,27]. It is a general and versatile model expressed as Equations (6) and (7):(6)$$Q_{t}=K_{f}C_{e}^{\frac{1}{n}},$$
(7)$$\ln Q_{t}=\left(1/n\right)\ln C_{e}+\ln K_{f},$$
where *Q_t_* is the apparent adsorption capacity (mg·g^−^^1^), *Q_e_* is the saturated adsorption capacity (mg·g^−1^), *a* is the Langmuir adsorption coefficient (L·g^−1^), *C_e_* is the adsorption balanced concentration (mg·L^−1^), and *K_f_* and *n* are the adsorption coefficients.

Gibbs free energy change (**∆***G**^θ^*), the adsorption enthalpy change (**∆***H^θ^*), and the adsorption entropy change (**∆***S^θ^*) are calculated to study the adsorption thermodynamics [28,29]. The calculation of adsorption equilibrium constant *K* and the relationship among the three thermodynamic parameters are shown in Equations (8)–(11).
(8)$$K=M_{w}C_{B}b,$$
(9)$$△G^{\theta }=△H^{\theta }-T△S^{\theta }=-RT\ln K,$$
(10)$$\ln K=\left(△S^{\theta }/R\right)-\left(△H^{\theta }/RT\right),$$
(11)$$△S^{\theta }=\left(△H^{\theta }-△G^{\theta }\right)/T,$$

### 3.6. Adsorption Kinetics

The oscillating adsorption method was adopted to determine the adsorption kinetic curves of 2700 on the surfaces of two crystals of pyraclostrobin particles [30,31]. Briefly, a certain amount of pyraclostrobin was placed in a 100 mL flask. Then, a certain concentration of 2700 solution was added. The flasks were sealed and placed at 298.15, 308.15, 318.15 K under oscillation condition. At the same time, a blank experiment was performed to eliminate the interference of pyraclostrobin in the water on UV absorption. The supernatant was sampled at regular intervals, and its concentration was determined following the method in Section 3.5.

To investigate the kinetic constants of adsorption, the data of adsorption amount at different time are fitted by pseudo-first-order kinetic model Equation (12) and pseudo-second-order kinetic model Equation (13) [32].
(12)$$\ln\left(Q_{e}-Q_{t}\right)=\ln Q_{e}-k_{1}t,$$
(13)$$t/Q_{t}=1/k_{2}Q_{e}^{2}+t/Q_{e},$$
where *Q_t_* is the apparent adsorption capacity (mg·g^−1^), *Q_e_* is the saturated adsorption capacity (mg·g^−1^), *k*_1_ is the adsorption rate constants of pseudo-first order kinetic equation (min^−1^), *k*_2_ is the adsorption rate constants of pseudo-second order kinetic equation (g·mg^−1^·min^−1^), and *t* is the adsorption time.

The adsorption rate constant k satisfies the Arrhenius Equation (14) [33]:(14)$$\ln k=\ln z-E_{a}/RT,$$
where *E_a_* is the apparent activation energy (kJ·mol^−1^), *R* (8.314 J·K^−1^·mol^−1^) is the gas constant, *k* is the adsorption rate constant, T is the temperature of the solution (*K*), and *z* is the Arrhenius factor.

### 3.7. XPS Analysis

A certain amount of pyraclostrobin was placed in 100 mL flasks with a certain concentration of 2700 aqueous solution. Then, the suspensions were placed at 298.15, 308.15, and 318.15 K and oscillated for a certain period. After the suspensions were centrifuged, the precipitations were dried at a vacuum condition and then measured by XPS (Axis Ultra, Kratos, Stretford, Manchester, UK) [34,35].
(15)$$I_{d}=I_{0}*\exp\left[-d/\lambda \left(E_{k}\right)\right],$$
(16)$$\lambda \left(E_{k}\right)=49E_{k}^{-2}+0.11{\left(E_{k}\right)}^{0.5},$$
where *I*_0_ is the initial photoelectron intensity, *I*_d_ is the photoelectron intensity through thickness *d*, *d* is the adsorption thickness (nm), *E*_k_ is the photoelectron kinetic energy (ev), and *λ* is the average escape depth of the photoelectron (nm).

## 4. Conclusions

In this study, the adsorption properties of polycarboxylate dispersant 2700 on the surface of two different crystal pyraclostrobin particles were investigated. The crystal structure of two different crystal forms of pyraclostrobin before and after the adsorption of dispersant was characterized by XRD, SEM, and FTIR techniques. The crystal form of pyraclostrobin do not change after adsorption of dispersant.

The adsorption kinetics and thermodynamics studies of 2700 on the surfaces of different crystalline forms of pyraclostrobin particles show that the adsorption process of 2700 on the surfaces of pyraclostrobin crystal forms II and IV conform to pseudo-second-order kinetic adsorption model. The *E*a values for crystal forms II and IV are 12.93 and 14.39 kJ∙mol^−1^, respectively, which indicates that both adsorption processes are physical adsorption. The adsorption models of 2700 on the surfaces of pyraclostrobin crystal forms II and IV are in accordance with Langmuir adsorption isotherms and the adsorption processes are spontaneous and accompanied by entropy increase. The absolute values of the adsorption enthalpy change of pyraclostrobin crystal forms II and IV are both lower than 20 kJ∙mol^−1^, which means that the main adsorption force of 2700 on the surfaces of two pyraclostrobin crystal forms belongs to van der Waals force.

The adsorption process of crystal form II is an endothermic process. The adsorption equilibrium concentration of 2700 on the surface of pyraclostrobin crystal form II continues to increase as temperature rises. The adsorption process of crystal form IV is an exothermic process. The adsorption equilibrium concentration of 2700 on the surface of pyraclostrobin crystal form IV decreases continuously with the rise in temperature. In the case of a certain amount of polycarboxylate dispersion, pyraclostrobin crystal form IV easily reaches the adsorption equilibrium. Thus, it is more suitable for the suspension system of pyraclostrobin, which is different from the two crystal forms of pyraclostrobin. The results of studies on the stability of the pyraclostrobin suspension system are consistent.

The results of this study provide an important theoretical basis for the selection of polycarboxylate dispersants in the suspension concentrates of pyraclostrobin. During suspension, the amount of polycarboxylate dispersant should vary according to the crystal form of pyraclostrobin. This study also provides a reference for the research of polycrystalline pesticide suspension.

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
