# Peer review of "Adsorption Properties of Comb-Shaped Polycarboxylate Dispersant onto Different Crystal Pyraclostrobin Particle Surfaces"

_molecules, 2020, doi:10.3390/molecules25235637_

Round 1

Reviewer 1 Report

The manuscript reports interesting data on adsorption of polycarbonate dispersant on pyraclostrobin particle surfaces. The manuscript is well written and well organized. It is based on clear and well described approach and methodology. Experimental procedures are well described. The quality of Figures is good. The adsorption mechanism is well described. Langmuir and Freundlich adsorption equations were used for the analysis of adsorption kinetics data. Major conclusions are well supported by X-ray diffraction, XPS, SEM, FTIR data. The manuscript can be accepted in as-submitted form  

Reviewer 2 Report

The article is interesting, well written, and certainly deserves publication. Few comments for improvement:

  • Please include specific results in the abstract
  • Please include the adsorption models in the isotherms figures, i.e., Figures 6 and 7 are not necessary. Same for kinetics.
  • Please employ recent models that describe better the isotherm behavior. Take a look at the SLE or modified BET model.
  • A schematic illustration of the main adsorption mechanisms would be helpful
  • Please elaborate on the isotherm type according to IUPAC
